# A Cross Sectional Sampling Reveals Novel Coronaviruses in Bat Populations of Georgia

**DOI:** 10.3390/v14010072

**Published:** 2021-12-31

**Authors:** Lela Urushadze, George Babuadze, Mang Shi, Luis E. Escobar, Matthew R. Mauldin, Ioseb Natradeze, Ann Machablishvili, Tamar Kutateladze, Paata Imnadze, Yoshinori Nakazawa, Andres Velasco-Villa

**Affiliations:** 1National Center for Disease Control and Public Health, Tbilisi 0198, Georgia; lelincdc@gmail.com (L.U.); gbabuadze@gmail.com (G.B.); A.Machablishvili@ncdc.ge (A.M.); tamar_kutateladze@yahoo.com (T.K.); pimnadze@ncdc.ge (P.I.); 2Biological Sciences Platform, Sunnybrook Research Institute, Sunnybrook Health Sciences Centre, Main Campus, University of Toronto, Toronto, ON M4N 3M5, Canada; 3Centre for Infection and Immunity Studies, School of Medicine, Sun Yat-Sen University, Guangzhou 510080, China; shim23@mail.sysu.edu.cn; 4Department of Fish and Wildlife Conservation, Virginia Polytechnic Institute and State University, Blacksburg, VA 24601, USA; escobar1@vt.edu; 5Centers for Disease Control and Prevention, 1600 Clifton Rd. NE, Atlanta, GA 30333, USA; yik5@cdc.gov (M.R.M.); inp7@cdc.gov (Y.N.); 6Institute of Zoology, Campus S, Ilia State University, Tbilisi 0162, Georgia; ioseb.natradze@iliauni.edu.ge; 7Department of Public Health and Epidemiology, Faculty of Medicine, Main Campus, Ivane Javakhishvili Tbilisi State University, Tbilisi 0179, Georgia

**Keywords:** coronavirus, bats, georgia, eastern europe, phylogeny, alphacoronavirus, betacoronavirus

## Abstract

Mammal-associated coronaviruses have a long evolutionary history across global bat populations, which makes them prone to be the most likely ancestral origins of coronavirus-associated epidemics and pandemics globally. Limited coronavirus research has occurred at the junction of Europe and Asia, thereby investigations in Georgia are critical to complete the coronavirus diversity map in the region. We conducted a cross-sectional coronavirus survey in bat populations at eight locations of Georgia, from July to October of 2014. We tested 188 anal swab samples, remains of previous pathogen discovery studies, for the presence of coronaviruses using end-point pan-coronavirus RT-PCR assays. Samples positive for a 440 bp amplicon were Sanger sequenced to infer coronavirus subgenus or species through phylogenetic reconstructions. Overall, we found a 24.5% positive rate, with 10.1% for *Alphacoronavirus* and 14.4% for *Betacoronavirus*. Albeit *R. euryale*, *R. ferrumequinum*, *M. blythii* and *M. emarginatus* were found infected with both CoV genera, we could not rule out CoV co-infection due to limitation of the sequencing method used and sample availability. Based on phylogenetic inferences and genetic distances at nucleotide and amino acid levels, we found one putative new subgenus and three new species of *Alphacoronavirus*, and two new species of *Betacoronavirus*.

## 1. Introduction

Coronavirus (CoV) infection in humans (e.g., 229E, NL63) and other mammals were first reported in the 1960s [1,2]. CoVs are highly pathogenic for livestock, pets, wild animals, as well as birds where they were first described in chickens in the early 1930s [3,4]. During the period 2002–2003, coronaviruses were recognized as zoonotic agents with high pandemic potential after an outbreak of severe acute respiratory syndrome (SARS) that caused 8096 cases and 774 deaths [5,6]. In 2012, the Middle East respiratory syndrome (MERS) emerged as a new public health concern for it spread rapidly to several countries around the globe (mainly the Arabian Peninsula and the Republic of Korea) and for showing a considerably higher mortality rate (35%) than SARS. As of 31 July 2021, MERS has been implicated in 2578 confirmed cases and 888 deaths [6]. The most recent pandemic involving a CoV, SARS-CoV-2 resulted in coronavirus disease 19 (COVID-19), began during 2019 and has caused more than 260 million cases and more than 5 million deaths, confirmed as of 10 December 2021 [7]. The imminent pandemic potential of CoVs has triggered a global hunt for the most likely wildlife reservoir hosts to gain a better understanding of the origins and evolutionary history of CoVs to predict future pandemics [8].

In 2018, the International Committee on Taxonomy of Viruses (ICTV) refined the classification of CoVs to re-organize a greater viral diversity based on newly discovered viruses [6]. Thus, CoVs are within the *Riboviria* realm that encompasses all viral families having an RNA-dependent RNA polymerase (RdRp), *Orthornavirae* kingdom, *Pisuviricota* phylum, *Pisoniviricetes* class, *Nidovirales* order, *Cornidovirineae* suborder, and *Coronaviridae* family. *Coronaviridae* was further separated into two subfamilies, *Letovirinae* primarily affecting amphibians, and *Orthocoronavirinae* affecting vertebrates. *Orthocoronavirinae* comprises four genera, *Alphaconoravirus*, *Betacoronavirus*, *Deltacoronavirus*, *Gammacoronavirus* [9]. Alphacoronaviruses (AlphaCoVs) and betacoronaviruses (BetaCoVs) mainly affect mammals with their greatest health impact in livestock and humans, while deltacoronaviruses (DeltaCoVs) and gammacoronaviruses (GammaCoV) generally affect birds [6,9]. The family *Coronaviridae* currently recognizes 2 subfamilies, 5 genera, 26 subgenera and 46 species (https://talk.ictvonline.org/taxonomy/ accessed on 1 November 2021).

The reservoir host of SARS-CoV-2 has yet to be confirmed, it has been suggested that there may be cryptic circulation in an unidentified intermediate host species [5,8]. Notably, CoV genera of public and animal health concerns share common ancestries with CoVs clades circulating in bats around the world [10]. Phylogenetic reconstructions suggest that bats may have the longest evolutionary history with CoVs justifying an ecological bat-focused investigation screening for CoVs with pandemic potential [6,11]. Although several studies have examined the presence of CoVs in bats throughout Europe and Asia [12,13], there is still very limited information about bat-CoV diversity in the South Caucasus region [14]. Georgia spans over 69,700 km^2^, with diverse habitats and climates, including coastline to the west, high peaked mountains to the north, rich grasslands to the south, and more arid regions to the southeast, making it an ideal place to sample multiple environments of the South Caucasus.

Given that Georgia is located on the crossroads of Europe, Asia and the Middle East, the country is expected to share bat species and viruses of these three regions [15,16]. Over the last decade, Georgia’s National Centers for Disease Control has been conducting investigations to search for close relatives of a broad spectrum of pathogens of public health concern potentially circulating in bat populations of this country namely, *Leishmania* spp., *Bartonella* spp., *Brucella* spp., *Leptospira* spp., *Yersinia* spp., *Lyssavirus* spp., and Hantavirus [17,18,19]. Thus, we conducted a cross-sectional retrospective survey on CoVs in bats across Georgia to fulfill a gap in the understanding of the evolution, ecology and biogeography of CoVs.

## 2. Materials and Methods

### 2.1. Sampling

Bats were collected from July to October 2014 at eight locations within four regions of Georgia (Table 1) to look for close relatives of pathogens causing bacterial, parasitic or viral diseases of public health concern in the region. Animal experiments described in this study were performed in compliance with the Ministry of Environment and Natural Resources Protection of Georgia, special permission #4001, 18 July 2014, in accordance with Animal Care Ethics Committee at the National Center For Disease Control and Public Health of Georgia. Animal handling techniques followed CDC IACUC protocol #2096FRAMULX-A3.

Mist nets and hand nets were utilized depending upon roost type (e.g., caves, buildings, attics). The number of individuals caught per roost and per species was authorized with anticipation by the Ministry of Environment and Natural Resources Protection of Georgia. Bats were individually placed in cotton bags to be transported to the laboratory the day after capture.

Bats were euthanized in a BSL2+ laboratory facility at the NCDC-Lugar Center. Specimens, were subsequently, measured (weight and length), sexed, and morphologically inspected for species identification [20]. Rectal and oral swabs, as well as tissues such as brain, liver, lung, spleen and intestines were collected and stored at −80 °C for subsequent testing. Rectal swabs were the only specimens available to search for coronaviruses for this retrospective study. All tissues were committed to search for other pathogens such as *Leshmania* spp., *Yersinia* spp., *Leptospira* spp., *Bartonella* spp., *Brucella* spp., *Lyssavirus* spp., Hantavirus [17,18,19].

### 2.2. RNA Extraction and PCR Amplification

A total of 188 rectal swabs diluted in 1 mL of virus transport media (VTM) were vortexed and spun down in a refrigerated microcentrifuge for 5 min at 10,000 rpm. For total RNA extraction, 140 µL of the VTM homogenate were placed in a new tube with 560 µL of lysis buffer from the QIAamp Viral RNA Mini Kit (Qiagen, Germantown, MD, USA) according to the manufacturer’s instructions. All 188 total RNA extracts were screened with a one-step RT-PCR method described previously that uses degenerated primers targeting a relatively conserved 180 bp region within the RNA-dependent RNA polymerase (RdRp) gene [21]. A second 440 bp non-overlapping fragment was amplified as described by Lelli et al. [21], which together with the 180 bp product would generate an approximately 600 bp sequence that can be used for phylogenetic characterization to infer, genus, subgenus and species with greater confidence [6,9,11,21]. Amplicons were gel purified using QIAGEN MiniElute^®^ Gel Extraction Kit (250) and subsequently cycle sequenced using the BigDye Terminator Kit, version 3.1 (Applied Biosystems, Foster City, CA, USA) to obtain a consensus sequence. Cycle sequencing products were purified by precipitation with ice-cold 100% isopropanol once, and then with 70% isopropanol. Sequencing was conducted using an Applied Biosystems ABI Prism 3130 XL sequencer with data collection software version 4.0.

### 2.3. Sequence Analyses and Phylogenetic Reconstructions

Excellent quality sequences were obtained in only 9 out of the 46 positive hits (20%) for the 180 bp RT-PCR screening amplicon. Thus, consistent concatenation with their 440 bp counterparts to generate a ~600 bp sequence was not possible. Nonetheless, we were able to obtain high-quality sequences ranging from 385 to 440 bp in all 46 CoV positive samples using the 440 bp RT-PCR amplicon. Thus, we trimmed all 46 sequences to 411 bp that corresponded to the longest sequence length we could obtain with GenBank accession numbers OL791325 to OL791370. This 411 bp fragment was used to search for homolog sequences ranging from 65 to 98% nucleotide identity, with 100% query coverage, in the GenBank database using the nucleotide-NCBI-BLAST and MOLEBLAST tools [22]. Gathered GenBank reference sequences were then used to reconstruct preliminary phylogenetic trees for initial genus-level identification (data not shown). Moreover, we used review articles [6,10,11,23,24,25,26,27] to identify CoV reference sequences representing all currently ICTV recognized subgenera and species pertaining to *Alphacoronavirus* and *Betacoronavirus* [9]. We retrieved 164 reference sequences encompassing 14 subgenera and 19 species pertaining to *Alphacoronavirus*, and 106 reference sequences comprising 5 subgenera and 14 species within *Betacoronavirus*. All sequences were aligned with the 46 Georgian CoV positive sequences using MUSCLE [28] and trimmed to 411 bp using BioEdit [29]. Taxa were subsequently divided into two datasets, one for BetaCoV that encompassed 106 BetaCoV reference sequences, 27 CoV sequences obtained from Georgian bats and 5 taxa representing some AlphaCov species, while the second data set comprised 164 AlphaCoV reference sequences, 19 sequences from Georgian bats and 4 BetaCoV taxa encompassing different species. MEGA7 [30], was used to determine the most appropriate model of molecular evolution for these combined datasets. The GTR + G + I substitution model was chosen, out of 24 models tested, for our phylogenetic reconstructions based on the Bayesian information criterion (BIC), Appendix A [30]. A Bayesian phylogenetic reconstruction using MrBayes V 3.2 software [31], was run on these two datasets with two independent runs under the GTR + G + I substitution model with 15 million generations, 4 Markov chains each run and the sampling for tree parameters every 1000 generations to assess branch support and/or calculation of Bayesian posterior probabilities. Using the pairwise alignment tool available in BioEdit [29], we calculated both the average nucleotide identity (ANI) and the average amino acid identity (AAI) by comparing one on one sequences of the Georgian CoV clusters (separately, from both the AlphaCoV and BetaCoV alignments) with taxa of their closest subgenera, species, as well as with unidentified closest CoVs neighbors observed in the phylogenetic reconstruction. The subgenera and species demarcation criterion suggested in the current literature were also used to assign Georgian CoV clusters to these taxonomical classification levels [6,9,11,23,27].

To better appreciate the extent of the geographic distribution of AlphaCoV and BetaCoV found in bat species of Georgia with those bat species harboring highly similar CoVs through Eurasia, we mapped their natural geographic distribution using species range data from the International Union for Conservation of Nature IUCN red list of threatened species [32,33,34,35,36,37,38,39,40,41,42,43,44]. Restricting ranges to the geography currently occupied by each species according to expert’s records (i.e., extant-resident). Scientific names and synonymies were corroborated following the standards of the Integrated Taxonomic Information System. Bat species were then grouped based on the genus of the coronavirus detected (i.e., AlphaCoV and BetaCoV). To create the maps, administrative boundaries were generated from maps downloaded from DIVA-GIS. Spatial data were handled and displayed using ArcGIS 10.8 (ESRI 2021 and R R Core Team 2021) [45,46].

## 3. Results

Of the 188 bat rectal swabs, 10% (19/188) were positive for AlphaCoV RNA and 14% (27/188) contained BetaCoV s RNA, with an overall positive rate of 24.5% (46/188), (Table 1 and Appendix A. Please see Appendix A). In regard to gender, we captured 56% females (106/188) and 44% males (82/188), (Appendix A). Approximately 29% (31/106) of tested females were positive for CoV RNA with 11.3% (12/106) containing AlphaCoV and 17.9% (19/106) BetaCoV, (Appendix A). Meanwhile, males had a 18.3% (15/82) positivity rate to CoV RNA, with 8.5% (7/82) to AlphaCoV and 9.8% (8/82) to BetaCoV (Appendix A).

A total of 183 bats were collected at 7 locations with different degrees of agricultural perturbation across Georgia, where we found a 25% (46/183) CoV positivity rate (Table 1). The eighth collection site at Saadamio cave, village Saadamio of Senaki municipality, presented an unperturbed landscape dominated by trees, shrubs embedded in a mosaic of herbaceous cover, where we only caught 5 CoV negative individuals (Table 1). The overall CoV positive rate for all eight collection sites was 24.5% (46/188), (Table 1, Figure 1).

Collected bats belong to two families *Rhinolophidae* (*n* = 84) that encompassed a single genus with three species namely, *Rhinolophus euryale* (*n* = 40), *R. ferrumequinum* (*n* = 39) and *R. blasii* (*n* = 5), and *Vespertilionidae* (*n* = 104) that comprised 5 genera with a total of 8 species (Table 2). The overall CoV positivity rates within the *Rhinolophidae* and *Vespertilionidae* were 25% and 24%, respectively (Table 2). However, *Rhinolophus* spp. presented higher CoV positive rates for BetaCoV (19%) than for AlphaCoV (6%) (with exception of *Rhinolophus blasii* that was negative for both CoV genera). In contrast, *Miniopterus schreibersii* exclusively presented AlphaCoV, while *Myotis blythii* (35% overall positivity rate, with 15% AlphaCoV and 19% BetaCoV) presented similar positivity rates to those observed in *Rhinolophidae* bats (Table 2, Figure 2 and Figure 3). The remaining *Vespertilionidae* species (*Eptesicus serotinus*, *Pipistrellus pygmaeus*, *Myotis mystacinus*, *M. alcathoe* and *Nyctalus leisleri*) were CoV negative along this transversal sampling (Table 2, Figure 1). Notably, CoVs were not detected in all bat species in which less than 10 individuals were collected. Albeit different individuals of the same species were found infected with AlphaCoV and BetaCoV within the same locations (Figure 1, Table 2), the sequencing method approach used could not rule out the potential presence of CoV co-infection among individuals. Geographically, the majority of CoV positive bats were collected in the Western part of Georgia in the regions of Imereti (32/46 = 71%) and Samegrelo (4/46 = 8.7%). The remaining 10 positive bats were collected in South-Eastern part of the Country including Kvemo Kartli (7/46 = 15.2%). Remarkably, only 0.5% (1/45) of CoV positive bats were collected in the Kakheti region that represents the Eastern part of Georgia (Figure 1, Appendix A).

The 19 Georgian AlphaCoVs grouped as two unclassified subgenera (*n* = 12 taxa), and as two unclassified species within two subgenera (*n* = 7), (Figure 2). The largest cluster encompassing 10 sequences (locate at the top of the tree, marked with a bar in color dark red) collected from cave Gliana, village Kumistavi of Tskaltubo municipality corresponded to Georgian bats of three species, *Myotis blythii* (*n* = 7), *Miniopterus schreibersii* (*n* = 2), and *Rhinolophus euryale* (*n* = 1). This cluster formed a monophyletic clade with unclassified CoV sequences found in *Myotis myotis* bats from Hungary, Germany, Spain and Italy presenting ANI and AAI values of 97.1% and 99.5%, respectively (Figure 2, and Appendix A). Other CoVs sequences found in a *Myotis daubentonii* bat from China, an unidentified bat from Korea and a *Myotis* sp. from Hong Kong were also identified as closely related to this cluster of Georgia sequences. However, they presented lower overall ANI and AAI values around 87.1% and 96.1%, respectively. At a more ancestral node (0.94 posterior probability) this cluster of unclassified AlphaCoV, mainly associated with vespertilionid bats from Europe and Asia shared a common ancestry with *Pedacovirus I*, *Pedacovirus* and *Colacovirus* subgenera with more distant overall ANI and AAI values around 77.6% and 86.6% (Figure 2).

Two sequences (CoV153 and CoV152) obtained from *Rhinolophus euryale* bats from cave Taroklde, village Zodi of Chiatura municipality, grouped monophyletically with high support with bat CoV circulating in *Rhinolophus blasii* from Bulgaria (97.6% ANI and 100% AAI) and *Rhinolophus ferrumequinum* from France (87.1% ANI and 99.3% AAI) and Italy (86.4% ANI and 99.3% AAI) (Figure 2, Appendix A).

We identified a Georgian CoV sequence (GE_CoV10) obtained from a *Myotis emarginatus* bat as sister taxon of *Rhinolophus ferrumequinum alphacoronavirus HuB-2013* (with an ANI of 94.7 and an AAI of 100%), which falls within the *Decacovirus* clade (Figure 2).

The fourth AlphaCo cluster (comprising 6 sequences) was the most diverse considering the number of bats species it contained, *Miniopterus schreibersii* (*n* = 3) from cave Gliana at village Kumistavi of Tskaltubo municipality, *Rhinolophus ferrumequinum* (*n* = 1) from the managed reserve at Gardabani, *R euryale* (*n* = 1) from cave Gliana at village Kumistavi of Tskaltubo municipality, and *Myotis blythii* (*n* = 1) from cave Gliana at Tshaltubo. This Georgia cluster was monophyletic with sequences pertaining to *Myotis ricketti alpha coronavirus Sax-2011* and *Miniopterus bat coronavirus HKU8* bat CoV, as well as with two CoV obtained from *Molossus rufus* bats from Brazil, all occupying the *Myotacovirus* clade (Appendix A, Figure 2). The closest relatives (albeit they grouped inside a polytomy with a low support value 0.57) to this cluster of Georgian bat CoVs were sequences obtained from *Myothis blythii* from Kazakhstan (91% ANI and 100% AAI), *Miniopterus schreibersii* (90% ANI and 99.3% AAI) from Luxemburg and Spain, and *Myotis emarginatus* (89.8% ANI and 97.8% AAI). These Georgian sequences, together with all other sequences inside the polytomy shared a common ancestor with CoV sequences obtained from *Molossus rufus* from Brazil (also pertaining to *Myotacovirus*), with ANI (77.8%) and AAI (83.2%) values observed (Figure 2).

Regarding the Georgian bat BetaCoV s, twenty-four sequences obtained from, *Rhinolophus euryale* (*n* = 10), *R. ferrumequinum* (*n* = 6), *Myotis blythii* (*n* = 7), *M. emarginatus* (*n* = 1) collected from David Gajeri (*n* = 1), Gardabani (*n* = 6), Tskaltubo (*n* = 13), and Chkhorotsku (*n* = 4) segregated within this genus and shared a common ancestor with CoV reference sequences pertaining to the *Sarbecovirus* and the Severe acute respiratory syndrome–related coronavirus. Interestingly, this group of Georgian bat CoVs presented ANI and AAI values of 96.9% and 97.81% with CoVs mainly found in *R. blasii*, *R. ferrumequinum*, and *R. hipossideros* from Bulgaria, Spain, France, Italy and Luxemburg. SARSCoV and SARSCoV-2 viruses obtained from *Rhinolophus* bats, civets and humans presented ANI values with this group of Georgian BetaCoV s of 88.6% (97.45% AAI) and 88.8% (95.62% AAI), respectively (Appendix A).

Similarly, we found three CoVs obtained from *Myotis blythii* collected at Tskaltubo that presented the highest average nucleotide and amino acid identities with a group of unclassified merbecoviruses obtained from Vespertilionid bats from Italy (*Eptesicus serotinus* and *Nyctalus noctula* 96% ANI and 97.8% AAI), China (*Myotis pequinius* 94.6% ANI, 95.6% AAI), and Finland (*Eptesicus nilssonii* 95.6% ANI and 97.8% AAI). This group of unclassified CoVs shared a common ancestry with the 4 species of the *Merbecovirus* (Figure 3). These three Georgian CoVs had the highest ANI (84.9%) and AAI (96%) with the *Middle East respiratory syndrome-related coronavirus* (Appendix A), which then decreased for the remaining three species in the subgenus as follows, *Pipistrellus bat coronavirus HKU5* (82% ANI, 90.5% AAI), *Hedgehog coronavirus 1* (81.3% ANI, 88.3% AAI), *Tylonycteris bat conoravirus HKU4* (78.8% ANI, 89% AAI). 

All CoV positive bat species that were found in Georgia presented an overlapping geographic distribution across most of southern Europe, a portion of North Africa and across central Asia. Only the distributions of *Myotis blythii* and *Rhinolophus ferrumequinum*’s extended to parts of Southeast Asia with the latter having the widest distribution in the region (Figure 4). *Myotis ricketti* (renamed as *Myotis pilosus*) whose natural distribution seems to be restricted to Southeast Asia appears to be the primary host of *Myotis ricketti alpha coronavirus Sax-2011*. However, close relatives of this AlphaCoV species were found circulating in Georgia in *Miniopterus schreibersii*, *Rhinolophus ferrumequinum* and *Myotis blythii,* suggesting that broadly distributed bats species such as *Rhinolophus ferrumequinum* and *Myotis blythii* could have dispersed this CoV species to Central Asia and Western Europe (Figure 4).

## 4. Discussion

The genetic characterization of mammal-associated CoV using a region spanning 411 bp within the RdRp was robust enough to identify novel CoVs circulating in bat populations across Georgia, albeit no reliable evolutionary relationships among subgenera and species could be established consistent with previous investigations using RdRp fragments shorter than 806 nucleotides [6,9,23,27]. Overall, our analyses based on phylogenetic inference, as well as considering the ANI and AAI criteria proposed recently [6,9,27], demonstrated the circulation of one putative unclassified subgenus closely related to the *Pedacovirus* holotypes and the *Colacovirus* subgenus. Three unclassified species two of which are within the *Decacovirus*, and the other one within the *Myotacovirus*. Additionally, we found two putative unclassified species within *Sarbecovirus* and *Merbecovirus*.

Wilkinson et al.; provided robust evidence that strong monophyletic groups demarcating a subgenus and species should have Bayesian posterior probabilities higher than 0.9 [6]. The genera level trees constructed for this investigation depicted with high support values all ICTV recognized subgenera and species and were consistent with previous investigations where robust analyses were conducted [6,9,11,27]. Interestingly, most of our Georgian CoV clades grouped consistently within CoVs associated with the same bat species or genus across Europe, Asia, Africa or the Americas, which supports the hypothesis that CoVs have strong epizootiological associations with their bat hosts spanning most of their natural geographic distributions [9,16,23,47,48,49,50,51,52]. In some instances, there were multiple bat species with the same viruses such as in the SARS-like clade where *Rhinolophus* spp. and *Myotis blythii* shared closely related viruses [11,51,53]. Similarly, two different clusters within AlphaCoVs, one that was embedded within *Myotacovirus* clade from Georgia was dominated by sequences obtained from *Miniopterus schreibersii* and *Rhinolophus* spp. However, this group contained a CoV species mainly associated with *Myotis ricketti*, while highly similar CoV sequences have been obtained from *Myotis blythii* in Kazakhstan and Italy and other parts of Europe, suggesting *Myotis* spp. could be the main reservoir hosts for this CoV [49,53,54,55]. Another Georgia clade predominantly containing sequences from *Myotis blythii*, may represent a yet unclassified *Myotis blythii* alphacoronavirus subgenus, which was found also infecting *Rhinolophus euryale* and *Miniopterus schreibersii* bats. Interestingly, the cross-sectional sampling we undertook could not identify an infection pattern among sedentary (*Rhinolophus ferrumequinum*, *R. euryale*, *R. blasii*, *Myotis emarginatus*, *M. alcathoe* and *Eptesicus serotinus*), sedentary or short-distance migrants (*M. blythii*, *M. mystacinus*), short-distance seasonal migrant (*Miniopterus schreibersii*), partially migrant (*Pipistrellus pygmaeus*) and long-distance migrant (*Nyctalus leisleri*) bat species, that would intuitively render seasonal and long-distance migrant bat species with a greater diversity of CoVs [39,40]. Although *Miniopterus schreibersii* bats have been found infected either with alpha or beta CoVs throughout Europe, they are predominantly infected with AlphaCoV in South East Asia [47,51,56], as we noted in Georgia. All together, these results indicate relatively frequent CoV spillover infections among bat species sharing roosts highlighting a broad susceptibility of multiple bats species to CoV infection that could potentially lead to host shifts [23,27]. Alternatively, these results may suggest that there might be multiple bat hosts species associated with the circulation maintenance of each viral genus acting as host-community complexes. Bat species with broad natural ranges could be playing a central role in the gradual dissemination of alpha and beta CoVs to other susceptible bats species across Eurasia, making CoV infection persistence and its dissemination dynamics a rather complex issue across bat populations of World.

Wilkinson and colleagues proposed an ANI threshold of 77.6% for different subgenera within alpha CoVs and 71.7 for BetaCoVs with high confidence levels [6]. Conversely, Geldenhuys and colleagues report pairwise amino acid average divergence thresholds (that we converted to identity values using the following formula, 100—divergence value = identity) that demarcate species to subgenus with AAI values lower than 92.4% and from subgenus to genus with AAI values lower than 85.3% [27]. Thereby, all our inferences to characterize Georgian CoV to subgenus and species levels fell into expected ranges [6,27].

Herein, we report for the first time the co-circulation of alpha and BetaCoVs in bat species belonging to *Vespertilionidae* and *Rhinolophidae* across Georgia, which constitutes the first report for the region [14]. Mammal-associated CoV seem ubiquitous in populations of the *Vespertilionidae* and *Rhinolophidae* bat families across the world [47,48,49,50,55]. Nonetheless, there are few countries within the Americas, Europe, Asia, Africa and Australia where both CoV genera co-circulate in native bat populations [47,48,49,50,51,52,53,54,55,57,58,59,60,61,62,63]. Likely, countries with co-circulation of both mammal-associated CoV genera, reflect an intensified surveillance in their bat populations, particularly after CoVs found in both bat families were predominantly associated with human acute respiratory syndrome epidemics and pandemics in the recent two decades [5,25,26,64].

Our results are not surprising since Georgia contains fauna native to both Europe and Asia and is relatively close to the Middle East [15]. Its geographic location at the middle of this great bio-diversity corridor, explains the outstanding diversity of CoV found, despite its relatively small territory (69,700 km^2^), as well as predicts a similar CoV richness for all the unexplored countries in the surrounding region [16]. We noticed that all Georgian bat species found positive with CoVs had broad natural geographic distributions across Eurasia, parts of Africa and the Americas. Strikingly, Georgian CoVs share recent common ancestry with CoV found circulating in other widely distributed bats species across Eurasia and the Americas. This empirical observation suggests mammal-associated CoVs may undergo complex transmission dynamics across sympatric bat communities, perhaps highlighting the susceptibility of multiple bat species to CoV infection. Recent modeling efforts have assessed the risk for the presence of bat-associated zoonoses globally [56]. Some of these results suggest Western Asia (especially the region comprising Georgia) is a hot spot for the presence of BetaCoV with pandemic potential, outside of South East Asia Western Europe and Central Asia [15,16].

The great CoV diversity found in single caves and the close proximity among positive locations suggests CoV co-infections are likely to occur in bat populations across the region. However, the sequencing approach we used and the amount of sample available hampered the detection of CoV co-infections in our sample set. In addition, the poor sequencing results obtained in the 180 bp pan-coronavirus RT-PCR could be due to a poor amplicon yield associated to low primer affinity (due to more than two sequence mismatches) along the highly variable region of the RdRp gene this 180 bp RT-PCR targets. This is clearly indicated by the 7 degenerated positions in the primer set used to amplify this 180 bp region [21]. Thus, this short 180 bp amplicon is practically useless for further CoV phylogenetic characterization. For future research we would only recommend the use of the 440 bp long amplicon (that anneals along a much more conserved region within the RdRp), which could be used for screening and phylogenetic characterization when a limited amount of sample is available [21]. Moreover, the sequencing of such an amplicon by next generation platforms may allow the detection of CoV co-infections in single bats.

## 5. Conclusions

CoVs found in Georgian bat populations were strikingly diverse. Some of these Georgian CoVs were closely related to those that caused human pandemics with epicenters in China (SARS) and the Middle East (MERS). We predominantly described new bat CoV species within previously described subgenera within *Alphacoronavirus* and *Betacoronavirus*, with one likely novel, yet unclassified, a subgenus of *Alphacoronavirus* associated with *Myotis blythii*. Our results provide further insight into the global genetic structure of CoV associated with bats in Eurasia and may inform the overall regional CoV diversity, critical to assess the global epidemiology of emerging diseases.

## Figures and Tables

**Figure 1 viruses-14-00072-f001:**
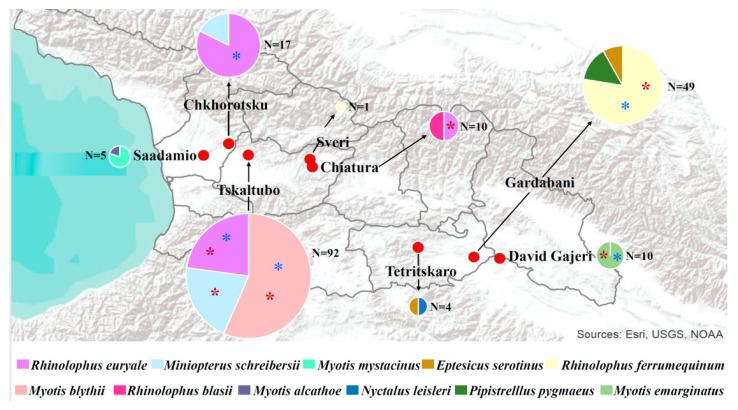
Geographic location and relative proportion of bat species collected per collection site. Red circles indicate the approximate location of collection site. Pie charts indicate the relative proportion of each bat species collected per site. Size of the pie charts indicates the relative number of samples collected per site. Asterisks within the pie indicate whether individuals of that species tested positive for alpha (red) and/or beta CoVs (blue). Lack of asterisks indicates no CoV were detected in a bat species and/or site. Color codes for each bat species represented in the pie charts are indicated in the lower part of the figure.

**Figure 2 viruses-14-00072-f002:**
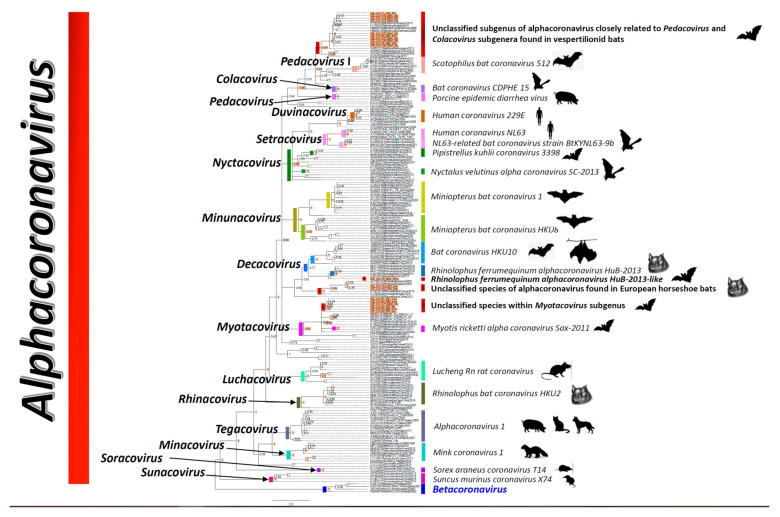
Unrooted Bayesian phylogenetic tree for AlphaCoVs generated with a partial informative 411 bp fragment of the RdRp. Representative BetaCoV taxa were used in the alignment to demonstrate consistent segregation between both genera. Values at nodes represent branch support values expressed as Bayesian posterior probabilities. Scale bar indicates branch lengths. Highlighted nodes are robustly supported with colored bars indicating relevant branches encompassing taxa pertaining to a given subgenus on the left. Bars on the right indicate taxa pertaining to recognized coronavirus species according to the most recent ICTV classification scheme. Taxa highlighted in orange on the right side of the tree indicate coronavirus sequences obtained from Georgian bats. Dark red bars indicate unclassified coronavirus subgenera or species identified herein. Taxa names for all reference sequences in the tree start with their respective GenBank accession number. Animal silhouettes on the extreme right indicate the animal species from which reference sequences were recovered.

**Figure 3 viruses-14-00072-f003:**
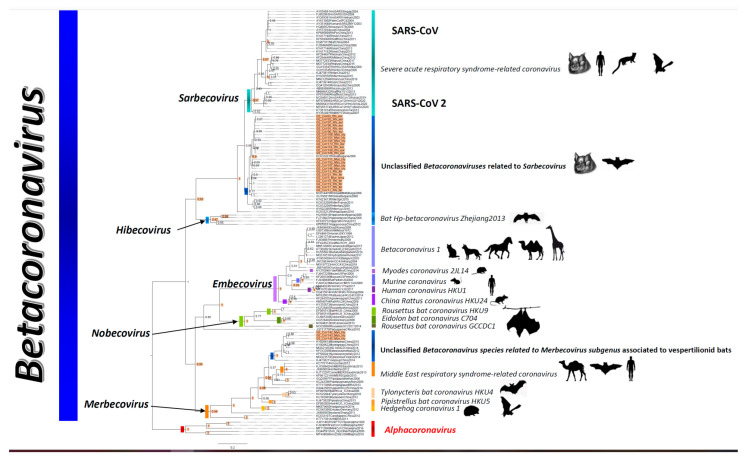
Unrooted Bayesian phylogenetic tree for beta CoVs generated with a partial informative 411 bp fragment of the RdRp. Representative BetaCoV taxa were used in the alignment to demonstrate consistent segregation between both genera. Values at nodes represent branch support values expressed as Bayesian posterior probabilities. Scale bar indicates branch lengths. Highlighted nodes are robustly supported with colored bars indicating relevant branches encompassing taxa pertaining to a given subgenus on the left. Bars on the right indicate taxa pertaining to recognized CoV species according to the most recent ICTV classification scheme. Taxa highlighted in orange on the right side of the tree indicate CoV sequences obtained from Georgian bats, and the dark blue bars indicate unclassified CoV species. Taxa names for all reference sequences in the tree start with their respective GenBank accession number. Animal silhouettes on the extreme right indicate the animal species from which reference sequences were recovered.

**Figure 4 viruses-14-00072-f004:**
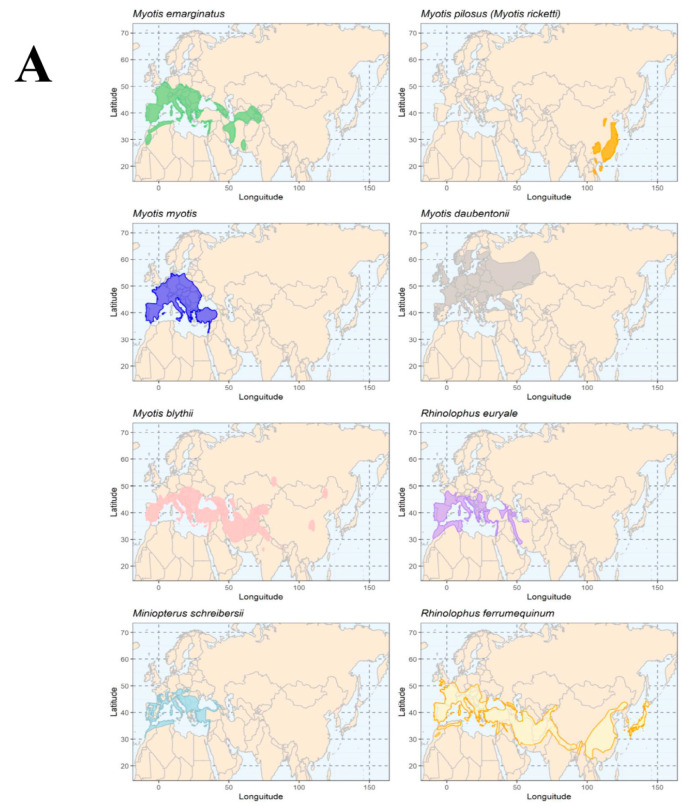
Distribution maps for bat species found to be CoV positive in this study compared with the geographic range of other bat species presenting highly similar CoVs across Eurasia as observed in our phylogenetic reconstructions. (**A**) depicts the geographic range of bat species with highly similar AlphaCoV obtained from Georgia with those reported in other countries across Eurasia. (**B**) depicts the geographic range of bat species with highly similar BetaCoV obtained from Georgia with those reported in other countries across Eurasia.

**Table 1 viruses-14-00072-t001:** Detailed location of collection sites and type of land use or cover. Overall and per genus coronavirus positive rate.

Location	Coordinates Long, Lat/Habitat	Tested	Positive	Overall CoV Positive Rate (%)	Positive Rate Per CoV Genus
David Gajeri, Tetri Senakebi	41.536, 45.257 Mostly natural vegetation in a mosaic with cropland	10	2	20	α = 10% *n* = 1 β = 10% *n* = 1 Neg = 80% *n* = 8
Gardabani Managed Reserve	41.376, 45.079 Mostly natural vegetation in a mosaic with cropland	49	7	14	α = 2% *n* = 1 β = 12% *n* = 6 Neg = 86% *n* = 43
Tskaltubo, cave Gliana	42.373, 42.597 Mostly cropland in a mosaic with natural vegetation	92	31	34	α = 16.5% *n* = 15 β = 17.5% *n* = 16 Neg = 66% *n* = 60
Chiatura.Taroklde cave	42.345, 43.308 Rainfed cropland	10	2	20	α = 20% *n* = 2 Neg = 66.7% *n* = 4
Chkhorotsku. Cave Lescurcume	42.529 42.102 Mostly cropland in a mosaic with natural vegetation	17	4	24	β = 24% *n* = 19% Neg = 81% *n* = 17
Saadamio Senaki	42.324, 42.103 Mostly trees and shrubs in a mosaic of herbaceous cover	5	0	0	Neg = 100% *n* = 5
Tetritskaro, Sabneleti	41.581, 44.582 Rainfed cropland	4	0	0	Neg = 100% *n* = 4
Sveri kvabkari	42.224, 43.302 Rainfed cropland	1	0	0	Neg = 100% *n* = 1

Natural vegetation = trees, shrubs and herbaceous cover.

**Table 2 viruses-14-00072-t002:** Overall and per genus coronavirus positive rate across bat species collected.

Species	Tested	Overall CoV Positive Rate (%)	Positive Rate Per CoV Genus
*Rhinolophus euryale*	40	14/40 = 35	α = 4/40 = 10 β = 10/40 = 25 Negative = 26/40 = 65
*Rhinolophus ferrumequinum*	39	7/39 = 18	α = 1/39 = 3 β = 6/39 = 15 Negative = 32/39 = 82
*Rhinolopus blasii*	5	0	Negative = 5/5 = 100
*Eptesicus serotinus*	6	0	Negative = 6/6 = 100
*Miniopterus schreibersii*	22	5/22 = 23	α = 5/22 = 23 Negative = 17/22 = 77
*Myotis blythii*	52	18/52 = 35	α = 8/52 = 15 β = 10/52 = 19 Negative = 34/52 = 66
*Myotis emarginatus*	10	2/10 = 20	α = 1/10 = 10 β = 1/10 = 10 Negative = 8/10 = 80
*Pipistrellus pygmaeus*	7	0	Negative = 7/7 = 100
*Myotis mystacinus*	4	0	Negative = 4/4 = 100
*Nyctalus leisleri*	2	0	Negative = 2/2 = 100
*Myotis alcathoe*	1	0	Negative = 1/1 = 100

## Data Availability

The data that have been used in this study will be available from the corresponding author upon requests.

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
