# Peer review of "A Cross Sectional Sampling Reveals Novel Coronaviruses in Bat Populations of Georgia"

_viruses, 2021, doi:10.3390/v14010072_

Round 1
Reviewer 1 Report
A well written manuscript describing the sampling of a number of bat species from Georgia; the subsequent testing for Coronaviruses and phylogenetic analysis of the positive samples. The authors identified 24.5% of bats were positive for coronaviruses, which include potentially new species. The link to the current Covid-19 pandemic makes this study relevant, and although the sequences obtained were not highly related to SARS-CoV-2, the results still are important to improve the understanding of the diversity of coronaviruses circulating in bat species, and the geographic range of these viruses. We should be encouraging all scientist to test archived bat samples for the presence of coronaviruses, as far too many samples sit within freezers for decades without being retested. The direct ancestor to SARS-CoV-2 could be in a freezer right now!
Major Comments:
- Please can you add some context to the study. The sampling was undertaken in 2014 and includes euthanising the bats. Although I appreciate there was ethical approval, this study only utilised the faecal swabs, so why were the bats euthanised? One would presume that the sampling was for a different pathogen/s requiring specific organs for testing, such as lyssaviruses. Certainly, sampling in 2014 couldn’t have been for SARS-CoV-2! Please can you update the manuscript to clarify so that readers are clear of the aims and ethical approval for collecting the samples in 2014. If the bats were sampled in 2014 for one pathogen and the samples have been subsequently used to look for SARS-CoV-2, this is perfectly valid, as long as the Ethical approval allows for this; and indeed should be promoted (see my overall comment above).
- Please add a statement that none of the bats had more than coronavirus detected. Has anyone found coinfections with coronaviruses? It always interests me when % positive is this high whether co-infections occur.
- Pg 4. Please discuss the reason why you used 2 PCRs and why one was so poor in yielding good sequences. My suspicion is that its so small (180bp – would only get ~100bp back from Sanger sequencing) but is it anything to do with primer binding e.g were the failures linked to the sequence itself). Why bother sequencing the 180bp product when you have a 600bp one? Some narrative around this would be appreciated.
- Please consider presenting data in Table 1 and 2 as graphs indicating positivity (the first 2 columns could be moved into supp material as its not essential for the paper).
Minor Comments:
- Abstract: please can you review this and improve its accuracy, particularly towards the end of the abstract from line 26 inwards?
E.g line 26 change to ‘Positive amplicons were sequenced…’
- line 36: remove ‘Nevertheless’
- Line 55: remove capitalised R in ‘realm’
- Line 65-68: reword this sentence for example: ‘The reservoir of SARS-CoV-2 has yet to be confirmed, it has been suggested that there is cryptic circulation in an unidentified intermediate host species {5,7].’
- Line 74: replace ‘has an area of’ with ‘spans’ or another suitable term
- Line 75: replace ‘beaches’ with ‘coastline’
- Line 137-8: Where is the output from MEGA to determine the appropriate model? Please add as a table.
- Line 211: spelling Myotis daubentonii
- Line 214-8: Please consider rewording this sentence
10: Figure 2 and 3 legends:
- Replace ‘A few’ with ‘Representative’
- Remove ‘at the bottom’
- Replace ‘Highlighted nodes with robust support are in yellow/orange’ with ‘Highlighted nodes are robustly supported’
- Line 252: resolve ‘and However’
- Line 357: replace ‘across bat populations of the world.’ With ‘across global bat populations.’
Author Response
Dear Reviewer,
We appreciate your assertive comments and revisions. They certainly contributed to improve the quality, completeness, and coherence of our investigation. Below, please find details on how we implemented your comments and suggestions.
A well written manuscript describing the sampling of a number of bat species from Georgia; the subsequent testing for Coronaviruses and phylogenetic analysis of the positive samples. The authors identified 24.5% of bats were positive for coronaviruses, which include potentially new species. The link to the current Covid-19 pandemic makes this study relevant, and although the sequences obtained were not highly related to SARS-CoV-2, the results still are important to improve the understanding of the diversity of coronaviruses circulating in bat species, and the geographic range of these viruses. We should be encouraging all scientist to test archived bat samples for the presence of coronaviruses, as far too many samples sit within freezers for decades without being retested. The direct ancestor to SARS-CoV-2 could be in a freezer right now!
Major Comments:
- Please can you add some context to the study. The sampling was undertaken in 2014 and includes euthanising the bats. Although I appreciate there was ethical approval, this study only utilised the faecal swabs, so why were the bats euthanised? One would presume that the sampling was for a different pathogen/s requiring specific organs for testing, such as lyssaviruses. Certainly, sampling in 2014 couldn’t have been for SARS-CoV-2! Please can you update the manuscript to clarify so that readers are clear of the aims and ethical approval for collecting the samples in 2014. If the bats were sampled in 2014 for one pathogen and the samples have been subsequently used to look for SARS-CoV-2, this is perfectly valid, as long as the Ethical approval allows for this; and indeed should be promoted (see my overall comment above). The text was modified accordingly along lines 100-103 in the introduction section and lines 184-187 in the material and methods section. Briefly, these specimens were long-termed stored remains of a more thorough sampling meant to look for close relatives of several human pathogens potentially circulating in bat populations of Georgia.
- Please add a statement that none of the bats had more than coronavirus detected. Has anyone found coinfections with coronaviruses? It always interests me when % positive is this high whether co-infections occur. We add such a statement in lines 621-623 at the results section, as well as discussed it as potential limitations of our study along lines 1226-1229 in the discussion section. Particularly, we, clarify limitations of our experimental design and sample volume availability that may have hampered ruling out CoV co-infection in individual bats. In addition, we offered some potential solutions to overcome such a limitations in future research, lines 1238-1239.
- Pg 4. Please discuss the reason why you used 2 PCRs and why one was so poor in yielding good sequences. My suspicion is that its so small (180bp – would only get ~100bp back from Sanger sequencing) but is it anything to do with primer binding e.g were the failures linked to the sequence itself). Why bother sequencing the 180bp product when you have a 600bp one? Some narrative around this would be appreciated. We added some statements in this regard along lines 1229-1236 in the discussion section where we talk about study limitations and future recommendations.
- Please consider presenting data in Table 1 and 2 as graphs indicating positivity (the first 2 columns could be moved into supp material as its not essential for the paper). We tried to balance the number of figures and tables across the main body of the manuscript. Thus, we decided to keep both tables, removing redundant or unnecessary information, to complement the information depicted in Figure 1
Minor Comments:
- Abstract: please can you review this and improve its accuracy, particularly towards the end of the abstract from line 26 inwards? The abstract was rewritten accordingly, trying to improve completeness and accuracy, lines 19-33.
E.g line 26 change to ‘Positive amplicons were sequenced…’Removed
- line 36: remove ‘Nevertheless’ words removed
- Line 55: remove capitalised R in ‘realm’ mistake corrected, line 75
- Line 65-68: reword this sentence for example: ‘The reservoir of SARS-CoV-2 has yet to be confirmed, it has been suggested that there is cryptic circulation in an unidentified intermediate host species {5,7].’This sentence was reworded line 86-87
- Line 74: replace ‘has an area of’ with ‘spans’ or another suitable term term substituted with the word suggested line 94
- Line 75: replace ‘beaches’ with ‘coastline’ word substituted as suggested line 95
- Line 137-8: Where is the output from MEGA to determine the appropriate model? Please add as a table. We added a supplementary table (now Table S1) with all 24 substitution models tested. We also added some verbiage in this regard across lines 369-371 in the material and methods section
- Line 211: spelling Myotis daubentoniid corrected now line 748
- Line 214-8: Please consider rewording this sentence Corrected accordingly
10: Figure 2 and 3 legends:
- Replace ‘A few’ with ‘Representative’ Corrected accordingly, line 765 and 842.
- Remove ‘at the bottom’ Corrected accordingly lines 767 and 844.
- Replace ‘Highlighted nodes with robust support are in yellow/orange’ with ‘Highlighted nodes are robustly supported’ Corrected accordingly, lines 767-768 and 844-845.
- Line 252: resolve ‘and However’ Corrected accordingly
- Line 357: replace ‘across bat populations of the world.’ With ‘across global bat populations.’ Corrected accordingly
Reviewer 2 Report
The manuscript describes the molecular identification of coronaviruses in bat samples collected in Georgia. The research subject is actual, significant and soundness and the report of new data from geographic regions here no information was previously available is extremely important to understand coronaviruses diversity.
Overall, the manuscript analysis a substantial amount of data, provides important information, and deserves to be published. However, is at some points confusing and difficult to follow. Most of online resources referred are incomplete and do not link to the specific information presented (as ICTV links).
A key issue is that most of the Figures presented are to small and impossible to read in detail in the presented size format. This is extremely important especially for Figure 2 and 3, since at current size, the trees that present the most important results of this work, should be clear in a way that the reader could identify the new bat coronaviruses identified in Georgia and all the viral diversity observed, it is impossible to confirm their phylogenetic position. Even in the download figure files the trees are impossible to check. A new format, using a bigger image in landscape or by adopting a coalescent viewing option of some clusters/viruses must be made (or any other possible option), to make the trees comprehensible. Although the methods follow to obtain the trees seem adequate, the obtained results are not available as presented, and this is a key issue that must be corrected.
I didn´t see the GenBank accession numbers for the new sequences presented in this study, not even in Table S1. The DNA and protein sequences accession numbers should be presented in Materials and Methods.
Results section description should be reviewed to be clearer and pinpoint the essential results. A suggestion would be to present ANI % and AAI % in a Table.
So, although the presented work is important, there are several points that need clarification and/or could be improved in the manuscript.
Specifically, I have some other comments and suggest the following:
- In the text, all referred online links should be numbered in square brackets [ ] and described in the References section; Lines 40, 45, 48, 59-60, 64, 156, 159
- Lines 54-59: CoVs taxonomic description is confusing. Since the realm is the highest taxonomic rank established for viruses, the taxonomic ranking goes up and down in the description and misses “Class” that would be helpful for an interested reader without expertise in this area. This description should be rewritten in a clearer way.
- Line 60-63: Acronyms of the four genera of coronaviruses are presented here but they appear in different formats through the text (capitalized, not capitalized, symbol…) . The choose presentation format should be uniform through the text, and if used different as in Table 1, referred in footnote.
- Table 1 : Land use and Habitat, as far as I can understand (in the presented form), provide identical information; I do not understand the importance of duplication that complicates the clarity of the table, without providing new significant information
- Table 1 (Column 2): there is no need to write Long and Lat for all entry lines if the column title refers that the first number is longitude and the second latitude. A clearer format would provide a clearer highlight of the important information.
- Lines 95-97: I would suggest changing to: “Specimens, were subsequently, measured (weight and length), sexed and morphological…”
- Line 97: please specify the tissues collected for subsequent testing.
- Line 101: correct microcentrifuge
- Lines 110-111: Why were the amplicons gel purified? The PCR amplification had multiple amplicons? Direct purification of PCR amplicons is more efficient and usually provides better sequencing results; probably why “excellent quality sequences were obtained only in 9 out of 46 positive hits (20%)” (line 117). I do not understand this methodologic option if the PCR "returns" a single amplification fragment…
- Lines 169,173, 174, 177, 271… : Numbers 1 to 9 should be written in full
- Lines 179, 331, 334, 336: Spp. and spp (in italic) should be corrected to spp.
- Line 174, 204: Tables and Figures references should not be within square brackets.
- Table 2: Remove / after Family (column 2)
- Figure 2 and 3: Substitute the images to enable visualization, and I suggest to change the legend to “ ...generated with a partial informative 410 bp fragment of RdRp …”
- Line 239: % symbol is missing at “…ANI of 97.7%…”
- Line 243: please correct euryale
- Lines 297-301: Rewrite this sentence: Maybe it misses “that” in “… suggesting that broadly distributed bats species…”?
- Line 342: Please correct: R. Euryale to R. euryale
- Lines 358 and 360: Wilkinson et al., 2020 and Geldenhuys et al., please remove date (line 358) and change one of the reference to Wilkinson and colleagues or Geldenhuys and colleagues.
- Line 425 (reference 3): this reference is incomplete, the manuscript title is missing;
right?
- References are not in agreement to Viruses Instructions and should be corrected.
- Manuscripts should follow:
Author 1, A.B.; Author 2, C.D. Title of the article. Abbreviated Journal Name Year, Volume, page range; DOI.
- References of Websites and online resources should be numbered in the text, and most of them, namely ICTV link are incomplete and in the presented way impossible to follow to access the referenced information, in the reference section should be stated as “Available online: http://... (accessed on date).”
Author Response
Dear Reviewer,
We appreciate your assertive comments and revisions. They certainly contributed to improve the quality, completeness, and coherence of our investigation. Below, please find details on how we implemented your comments and suggestions.
The manuscript describes the molecular identification of coronaviruses in bat samples collected in Georgia. The research subject is actual, significant and soundness and the report of new data from geographic regions here no information was previously available is extremely important to understand coronaviruses diversity.
Overall, the manuscript analysis a substantial amount of data, provides important information, and deserves to be published. However, is at some points confusing and difficult to follow. Most of online resources referred are incomplete and do not link to the specific information presented (as ICTV links). This is a very good point, and we apologize for the inconvenience. We added general links to the webpages where we obtained part of the information we are referring to, along our manuscript’s main body. However, in most instances as you asserted the link does not take you directly to the information, due to webpage configuration limitations. Unfortunately, the reader would have need to search for the information in the website as we did. A clear example of this limitation is the ICTV webpage in which we could not direct the link to the exact taxonomic categories where CoVs are located. Unfortunately, once one gets out to the webpage we had to look for the CoV location by scrolling down to the right place, all over again. Thus, to avoid any confusion from any potential reader we removed all weblinks with this configuration problems and only cited the most relevant references where readers may find such an information when possible.
A key issue is that most of the Figures presented are to small and impossible to read in detail in the presented size format. This is extremely important especially for Figure 2 and 3, since at current size, the trees that present the most important results of this work, should be clear in a way that the reader could identify the new bat coronaviruses identified in Georgia and all the viral diversity observed, it is impossible to confirm their phylogenetic position. Even in the download figure files the trees are impossible to check. A new format, using a bigger image in landscape or by adopting a coalescent viewing option of some clusters/viruses must be made (or any other possible option), to make the trees comprehensible. Although the methods follow to obtain the trees seem adequate, the obtained results are not available as presented, and this is a key issue that must be corrected. Thank you very much for this comment. We agree in that figures 1 to 4 are of critical importance for this study, therefore they require high level of resolution. We double checked the resolution quality of all our figures originally submitted to the journal. We noticed the pdf format we originally uploaded the figures to the journal’s submission platform, could be seen up to 800X with no loss of resolution. We are afraid that when the Journal’s production inserted the figures in the body of the text the images were inserted in a different format (jpg, tif or png), causing the figures to loss their original resolution. Thus, we contacted Viruses MDPI to make sure our figures would be downloadable in the original high resolution pdf format.
All figures were re-inserted as screen shots from the original high resolution pdf format. We noticed that figure’s details can be seen perfectly now.
I didn´t see the GenBank accession numbers for the new sequences pr.esented in this study, not even in Table S1. The DNA and protein sequences accession numbers should be presented in Materials and Methods. GenBank accession numbers OL791325 to OL791370 (which will describe nucleotide and amino acid sequences) were added in the material and methods section line 348, as well as along table S2, former table S1.
Results section description should be reviewed to be clearer and pinpoint the essential results. A suggestion would be to present ANI % and AAI % in a Table. We appreciate the suggestion. However, we tried several ways to depict such an information, but it was very challenging. As mentioned in your introductory review summary, we analyzed an outstanding amount of information. To be able to observe ANI and AAI values among sequences of particular interests to the level of detailed described in our results section, we have had to present ANI and AAI matrices with about 300 by 300 taxa for each CoV genera analyzed. Unfortunately, such matrices could not be displayed in an amenable table so that the reader could see taxa by taxa comparisons we showed in the results section. Nonetheless, we tried to create some readable matrices for you to evaluate. However, since we had to merged groups of related taxa, the level of detailed of the relevant comparisons reported along our results section was not no possible see. That is why we opted, as we explained in the material and methods section do the one on one comparisons of all the taxa of interest (nucleotide and amino acid pairwise alignment) using the pairwise alignment tool available in the BioEdit program, which only addressed the most relevant comparisons as depicted along the results section of this article.
Please check the draft matrices we created with MEGA 7 as a feasible alternative we included in our revised version. However, we are hesitant to include them in the final version of the manuscript because they do not depict with the same level of accuracy to the one-on-one comparisons, we reported in the results section using BioEdit. We suggest that the MEGA 7 draft ANI matrices could be made available, per request, or can be included as supplemental material with all caveats mentioned above. In our opinion, doing so. may be confusing for readers.
So, although the presented work is important, there are several points that need clarification and/or could be improved in the manuscript.
Specifically, I have some other comments and suggest the following:
- In the text, all referred online links should be numbered in square brackets [ ] and described in the References section; Lines 40, 45, 48, 59-60, 64, 156, 159. This is a very important point, as well, and we thank you for your remark. Please, see detailed response above where you first mentioned this problem.
- Lines 54-59: CoVs taxonomic description is confusing. Since the realm is the highest taxonomic rank established for viruses, the taxonomic ranking goes up and down in the description and misses “Class” that would be helpful for an interested reader without expertise in this area. This description should be rewritten in a clearer way. The complete taxonomic nomenclature for coronaviruses was corrected accordingly, by moving realm as the first hierarchical category and we also included the missing class name, lines 75-77
- Line 60-63: Acronyms of the four genera of coronaviruses are presented here but they appear in different formats through the text (capitalized, not capitalized, symbol…) . The choose presentation format should be uniform through the text, and if used different as in Table 1, referred in footnote. This is also a very good observation, thank you. Nomenclature was harmonized accordingly throughout the entire text, including figure’s footnotes, tables, etc..
- Table 1 : Land use and Habitat, as far as I can understand (in the presented form), provide identical information; I do not understand the importance of duplication that complicates the clarity of the table, without providing new significant information. We agree with the assertive comment, this was corrected accordingly redundant information was removed from tables
- Table 1 (Column 2): there is no need to write Long and Lat for all entry lines if the column title refers that the first number is longitude and the second latitude. A clearer format would provide a clearer highlight of the important information. This was corrected accordingly, we change the column and remove Long, Lat in all rows..
- Lines 95-97: I would suggest changing to: “Specimens, were subsequently, measured (weight and length), sexed and morphological…”This was corrected as suggested, please check line 181
- Line 97: please specify the tissues collected for subsequent testing. This information was added along lines 184-188
- Line 101: correct microcentrifuge mistake corrected line 190
- Lines 110-111: Why were the amplicons gel purified? The PCR amplification had multiple amplicons? Direct purification of PCR amplicons is more efficient and usually provides better sequencing results; probably why “excellent quality sequences were obtained only in 9 out of 46 positive hits (20%)” (line 117). I do not understand this methodologic option if the PCR "returns" a single amplification fragment…We appreciate the comment. We tried to implement the protocol as suggested in reference 21. We did not anticipate the potential limitation of using a primer set with so many degenerated positions. Some verbiage around this problem was discussed as limitations of our study, lines 1202-1208. We also offer some suggestions for future studies lines 1208-1212.
- Lines 169,173, 174, 177, 271… : Numbers 1 to 9 should be written in full We implemented your suggestion accordingly
- Lines 179, 331, 334, 336: Spp. and spp (in italic) should be corrected to spp. Corrected accordingly throughout the entire text.
- Line 174, 204: Tables and Figures references should not be within square brackets. Corrected accordingly throughout the entire text.
- Table 2: Remove / after Family (column 2) Corrected accordingly throughout the entire text.
- Figure 2 and 3: Substitute the images to enable visualization (please see comment above, we talked with the journal’s editorial office to request making available our original high resolution pdf figures as they were uploaded in our original submission), and I suggest to change the legend to “ ...generated with a partial informative 410 bp fragment of RdRp …” Corrected accordingly
- Line 239: % symbol is missing at “…ANI of 97.7%…” Corrected accordingly
- Line 243: please correct Euryale Mistake corrected accordingly.
- Lines 297-301: Rewrite this sentence: Maybe it misses “that” in “… suggesting that broadly distributed bats species…”? Corrected line 877 and lines 1186-1188.
- Line 342: Please correct: R. Euryale to R. euryale Corrected accordingly.
- Lines 358 and 360: Wilkinson et al., 2020 and Geldenhuys et al., please remove date (line 358) and change one of the reference to Wilkinson and colleagues or Geldenhuys and colleagues. Corrected accordingly. Lines 943 and 945, respectively
- Line 425 (reference 3): this reference is incomplete, the manuscript title is missing;
right? Corrected accordingly. Line 1309
- References are not in agreement to Viruses Instructions and should be corrected.
- Manuscripts should follow: Corrected accordingly. Thank you for the observation
Author 1, A.B.; Author 2, C.D. Title of the article. Abbreviated Journal Name Year, Volume, page range; DOI.
- References of Websites and online resources should be numbered in the text, and most of them, namely ICTV link are incomplete and in the presented way impossible to follow to access the referenced information, in the reference section should be stated as “Available online: http://... (accessed on date).” Corrected accordingly. Thank you for the observation, corrections can be seen at lines 1305 to 1626
Round 2
Reviewer 2 Report
I think that most of my comments were addressed and with the added changes, the manuscript was improved.
However, I still think that some minor points could be enhanced before publication:
Line 66- substitute ] by );
Lines 83, 83, 17-109, 194 and 354 – spp. please add dot;
Line 85 – please remove spp. after Hantavirus;
Lines 62, 274, 283 – remove space in AlphaCoV s and BetaCoV s to AlphaCoVs and BetaCoVs;
Lines 197, 198 _remove extra space after AlphaCoV and BetaCoV
Line 135 – add n to numbers
Regarding the inclusion of Table S1, I have nothing against it, but I think there is no need to add this information.
Author Response
I think that most of my comments were addressed and with the added changes, the manuscript was improved.
However, I still think that some minor points could be enhanced before publication:
Dear Revewer 2, On behalf all authors I would like to thank you very much for your corrections. We implemented all your assertive corrections accordingly.
Line 66- substitute ] by ); corrected accordingly
Lines 83, 83, 17-109, 194 and 354 – spp. please add dot; all corrected accordingly
Line 85 – please remove spp. after Hantavirus; corrected accordingly
Lines 62, 274, 283 – remove space in AlphaCoV s and BetaCoV s to AlphaCoVs and BetaCoVs; space removed accordingly
Lines 197, 198 _remove extra space after AlphaCoV and BetaCoV space removed
Line 135 – add n to numbers corrected
Regarding the inclusion of Table S1, I have nothing against it, but I think there is no need to add this information. We do agree with your observation, that is why we had not included in the first version submitted for your review. However, the inclussion of the results for substitution model analysis was a request from reviwer 1. Thus, if you do not mind we would like to include it.